# Air Quality Variation in Wuhan, Daegu, and Tokyo during the Explosive Outbreak of COVID-19 and Its Health Effects

**DOI:** 10.3390/ijerph17114119

**Published:** 2020-06-09

**Authors:** Chang-Jin Ma, Gong-Unn Kang

**Affiliations:** 1Department of Environmental Science, Fukuoka Women’s University, Fukuoka 813-8529, Japan; 2Department of Medical Administration, Wonkwang Health Science University, Iksan 54538, Korea; gukang@wu.ac.kr

**Keywords:** PM_2.5_, NO_2_, COVID-19, health effect, exposure dose, Wuhan, Daegu, Tokyo

## Abstract

This study was designed to assess the variation of the air quality actually measured from the air pollution monitoring stations (AQMS) in three cities (Wuhan, Daegu, and Tokyo), in Asian countries experiencing the explosive outbreak of COVID-19, in a short period of time. In addition, we made a new attempt to calculate the reduced *Dose_PM_*_2.5_ (μg) at the bronchiolar (Br.) and alveolar-interstitial (AI) regions of the 10-year-old children after the city lockdown/self-reflection of each city. A comparison of the average PM_2.5_ of a month before and after the lockdown (Wuhan) and self-reflection (Daegu and Tokyo) clearly shows that the PM_2.5_ concentration was decreased by 29.9, 20.9, and 3.6% in Wuhan, Daegu and Tokyo, respectively. Wuhan, Daegu and Tokyo also recorded 53.2, 19.0, and 10.4% falls of NO_2_ concentration, respectively. Wuhan, which had the largest decrease of PM_2.5_ concentration due to COVID-19, also marked the largest reduced *Dose_PM_*_2.5 *10-year-old children*_ (μg) (3660 μg at Br. and 6222 μg at AI), followed by Daegu (445 μg at Br. and 1287 μg at AI), and Tokyo (18 μg at Br. and 52 μg at AI), over two months after the city lockdown/self-reflection. Our results suggest that the city lockdown/self-reflection had the effect of lowering the concentration of PM_2.5_, resulting in an extension of the period it took to the acute allergic airway inflammation (AAI) for the 10-year-old children.

## 1. Introduction

Since the initial report of cases in Wuhan (see Figure 1), China, on 31 December 2019, the coronavirus disease 2019 (COVID-19) spread worldwide in a short period of time and is still in progress [1]. During the pandemic, 4,445,920 confirmed cases were reported in 213 countries and 298,440 people have died so far from the COVID-19 outbreak, as of 14 May 2020 [2].

In response to the rapid increase in the COVID-19 case, the Chinese administrative authorities have sealed off the entire city of Wuhan. As a concrete city blockade, they blocked traffic and banned people from moving out of town. To prevent the spread of COVID-19, they also closed various educational institutions and established numerous quarantines [3].

As shown in Figure 2, in the case of Daegu Metropolitan City, South Korea, the number of cases sharply increased at the time when the number of confirmed cases of Wuhan was on the decline. In Daegu, just 17 days after the first case was reported on 18 February, an explosive outbreak of COVID-19 occurred.

The authorities of Daegu also implemented various administrative regulations to prevent the outbreak of COVID-19, and the number of newly confirmed cases decreased to 5 cases per day on 5 April (see Figure 3).

Meanwhile, the number of confirmed cases increased in Tokyo following the sharp drop in Daegu, reaching 197 daily confirmed cases on 11 April (see Figure 2 drawn from the data of Toyo Keizai Online [4]). As COVID-19 spread rapidly, Tokyo also issued executive orders, such as temporary closure request for public schools (3 March), request for self-reflection at home (25 March), and declaration of the state of emergency (7 April) (see Figure 3).

As mentioned above, on 23 January 2020, the central government of China imposed a lockdown in Wuhan (see Figure 3) and other cities in Hubei, when the cumulative confirmed cases reached 1410. Daegu and Tokyo have requested self-reflection on 23 February and 25 March, when the cumulative confirmed cases were 319 and 212, respectively. Until the official city lockdown was lifted by 8 April, all vehicles and people in Wuhan were not allowed to move. There was no compulsion like Wuhan, but the number of vehicles and floating population dropped sharply after the request for self-reflection in Daegu and Tokyo, too.

Meanwhile, while people were facing COVID-19 worldwide, the unexpected positive effects of environmental pollution decreasing have been reported. Among them, the satellite imagery of NASA [5] has shown how NO_2_ has improved drastically in big cities across China during the COVID-19 pandemic.

According to a report from the World Health Organization (WHO) [6], the exposure to toxic air, both indoors and outdoors, kills some 600,000 children under the age of 15 each year. Brauer [7] also suggested that air pollution kills seven million people, especially for infants and the elderly, worldwide every year, and more than 90% of people around the world breathe polluted air. Even in European countries, 193,000 people died in 2012 because of airborne particulate matter [8].

It is well known that the exposure to particulate matter with a diameter of less than 2.5 μm (PM_2.5_) has been related to both acute and chronic respiratory diseases. Many epidemiologic studies have suggested that children are more vulnerable to PM_2.5_ [9,10].

According to the results of several previous studies, it can be said that air pollution causes far more human deaths than COVID-19 from a long-term perspective. It is quite meaningful to assess the improvement of air quality during the COVID-19 pandemic through actual measurement data and to evaluate the health effects of the improved amount, especially on children.

In this study, the air quality variation with the trend of COVID-19 at Wuhan in China, Daegu in South Korea, and Tokyo in Japan experienced explosive outbreaks in a short period of time, which was estimated based on the actual measured data from air pollution monitoring stations (AQMS). The health effect of the reduced PM_2.5_ dose due to COVID-19 on 10-year-old children in each city was also quantitatively assessed.

## 2. Methods

### 2.1. Sites Description

Wuhan is the capital city of Hubei province in China, with a population of over 11.08 million. It covers an area of around 8,494 km^2^ and has a humid subtropical climate, with abundant rainfall in summer and four distinctive seasons [11]. By the end of 2018, the total number of registered motor vehicles in Wuhan was 2.97 million, accompanied by increasing urban traffic pressure [12].

Daegu Metropolitan City, South Korea is the fourth-largest after Seoul, Busan, and Incheon. Its major industries are the textile industry, metals and machinery, the automobile component industry, mobile development, and medical care. As of May 2019, the population and area of this city are 2.49 million and 883.54 km^2^, respectively. The total number of cars registered was 1,190,154, of which gasoline, diesel, LPG, CNG and hybrid were 561,643, 470,569, 114,533, 2617, and 26,026, respectively [13].

Tokyo, the capital of Japan, is located at the head of Tokyo Bay on the Pacific coast of the central Honshu. This metropolitan area is the largest industrial, commercial, and financial center in Japan. The area and population of Tokyo are approximately 2188 km^2^ and 13.95 million, as of 1 January 2020 [14]. In 2019, a total of 3.95 million motor vehicles were registered in Tokyo. Total motor vehicles include cars, trucks and buses, as well as special purpose vehicles [15].

### 2.2. One-Hour Interval Measured Data and Those Sources

In this study, the data of PM_2.5_ and NO_2_ measured continuously at one-hour intervals at the air quality monitoring stations (AQMSs) of three cities were studied. The data monitored at the AQMSs of three cities from 9 January to 29 April 2020 became the subjects of this study.

The data of Wuhan were monitored at the Hankoujiangtan urban AQMS (30.59 N, 114.30 E) and published on the website of the Ministry of Environmental Protection (MEP) in China [16]. The average monthly temperature in Wuhan ranged between 5.1 and 17.3 °C from January to April, 2020. Those of relative humidity and wind speed during the same period are 58.8–78.4% and 2.38–2.97 m/s, respectively. The atmospheric environment standards of PM_2.5_ and NO_2_ in China are below 35 μg/m^3^ and 40 ppb, respectively, on annual average concentration.

The data of Daegu were monitored at the Suchang-dong AQMS (35.52 N, 128.36 E), installed in an urban area. The average monthly temperature (°C), relative humidity (%), wind speed (m/s) in Daegu during the measurement period ranged from 3.8–13.5 °C, 52–61%, and 2.1–2.4 m/s, respectively. The environmental quality standards in South Korea for the 24-h average and annual average of PM_2.5_ are <35 and <15 μg/m^3^, respectively. In the case of NO_2_, the daily average for hourly values shall be within the 40–60 ppb or below this zone.

The selected AQMS in Tokyo was Shinjuku (35.69 N, 139.70 E), a major commercial and administrative center for the government of Tokyo. Wind speed (m/s), temperature (°C), and relative humidity (%) at the monitoring sites of Shinjuku ranged from 0–2.9 m/s, 0.9–22.1 °C, and 21.8–97.7%, respectively. The environmental quality standards are the same as those of Korea.

### 2.3. Data Handling

To better represent the time series trend of PM_2.5_ concentration over the whole measurement period, all data were treated with the 5-day simple moving average (SMA) $\left({\bar{C}}_{\bar{d}\;SMA}\right)$ by the following equations:(1)$${\bar{C}}_{\bar{d}\;SMA}=\frac{C_{\bar{d}}+C_{\bar{d}-1}+\cdots +C_{\bar{d}-\left(n-1\right)}}{n}=\frac{1}{n}\overset{n-1}{\underset{i=0}{\sum }}C_{\bar{d}-i}$$
where $C_{\bar{d}}$ is the daily average of measured data every hour, and *n* is the number of days in an interval of daily average data (5-day in this study). The ${\bar{C}}_{\bar{d}\;SMA}$ can help us distinguish between typical measurement noise and actual trends.

## 3. Results and Discussion

### 3.1. Variation of PM_2.5_ and NO_2_ Concentration with Confirmed Cases of COVID-19

Figure 4 shows the time series variation of the concentration of PM_2.5_ in Wuhan, Daegu, and Tokyo, with their cumulative confirmed cases of COVID-19.

Unlike Tokyo, where the concentration was not high before the self-restraint regulation, there was a clear reduction in the PM_2.5_ concentration in Wuhan and Daegu. Although it was not a continuous reduction, the concentration of PM_2.5_ in Wuhan showed a significant decrease. It is necessary to assess whether this is a simple seasonal variation or a change due to the city lockdown. The results of our study are compared with those of Gong et al. [17], which investigated the monthly concentration of PM_2.5_ in Wuhan in 2015. In their study, the monthly average of PM_2.5_ from January to April was 146, 82, 87, and 72 μg/m^3^. Meanwhile, in this study, the concentrations of PM_2.5_ in each month during the same period were 80.8, 51.7, 52.0, and 48.8 μg/m^3^. Although our results are like the seasonal trends in 2015, the PM_2.5_ concentrations for each month from January to April were greatly reduced by 44.7, 37.0, 40.3, and 32.3%, compared to those of 2015. In February, when the number of the confirmed COVID-19 cases was at its peak, PM_2.5_ concentration was 57.3% lower than that in February 2017 (121.2 μg/m^3^) [18]. Therefore, it can be said clearly that the city lockdown has resulted in a great reduction in PM_2.5_.

In Daegu, the concentration of PM_2.5_ shows a temporary increase or decrease, but the trend of decline was evident in the overall period.

The overall decline of the PM_2.5_ in Tokyo is not seen, but it is clearly decreasing after request for self-reflection on 25 March.

Apart from the overall pattern of decline, several short reduction intervals due to rainfall were also clearly found in all three cities.

Figure 5 shows the result of comparing the average PM_2.5_ of a month before and after the lockdown (Wuhan) and self-reflection (Daegu and Tokyo) in each city. The decreasing rate of PM_2.5_ concentration in each city was 29.9, 20.9 and 3.6% in Wuhan, Daegu and Tokyo, respectively.

Xu et al. [19] reported that, during the defined 3-week lockdown period, Wuhan’s PM_2.5_ level went down 44% from 2019. In the case of Tokyo, the decreasing rate was significantly lower compared to the two cities, probably because of usual low PM_2.5_ concentration.

Figure 6 shows the daily variations of the concentration of NO_2_ in Wuhan, Daegu, and Tokyo, with the cumulative confirmed cases of COVID-19.

The concentration of NO_2_ in Wuhan significantly decreased while the city was in lockdown from 23 January to 8 April. NO_2_ is generally emitted from traffic and factories and is therefore a good indicator of human activity outside the home. A large peak appeared around 8 April, when the Wuhan lockdown officially ended, may be because of the comeback of a floating population and traffic amount. Although smaller than this, the peak also appeared in the distribution of PM_2.5_ (see Figure 4).

In the cases of Daegu and Tokyo, the continuous reduction in NO_2_ concentration was more evident than that of PM_2.5_. Although there were occasional temporary reductions in NO_2_ because of the wash-out, the overall decline might also be due to people’s efforts, such as the reduction of traffic volume and the partial shutdown of industrial facilities under autonomous self-reflection.

Figure 7 shows the result of comparing the average NO_2_ of a month before and after the city lockdown (Wuhan) and self-reflection (Daegu and Tokyo) in each city.

Wuhan, Daegu, and Tokyo recorded 53.2, 19.0, and 10.4% fall of NO_2_, respectively. Although it is a short-term decline, it can be said that this result is very meaningful for the citizens’ health of three cities, especially Wuhan citizens. The model calculation by Dutheil et al. [20] suggested that the reduction of NO_2_ in China due to COVID-19 epidemic during a time period of two months saved around 100,000 lives in China.

### 3.2. Exposure Assessment

Exposure to PM_2.5_ can cause lung function abnormalities including increased airflow obstruction and airway hyper responsiveness [21]. According to research done by Gauderman et al. [22], the short-term exposure of PM_2.5_ on chronic obstructive pulmonary disease demonstrated the impact of PM_2.5_ on years of life lost. Li et al. [23] suggested that the deficits in the growth of the expiratory volume in one second for the ages between 10 and 18 years old were associated with exposure to PM_2.5_. Airborne allergens usually develop by 10 years of age, and this reaches its peak in the teens or early twenties [24].

As mentioned above, the difference in the average concentration of PM_2.5_ in the month before and after self-reflection were 18.0, 5.3, and 0.36 µg/m^3^ in Wuhan, Daegu, and Tokyo, respectively. To find out what the reduced PM_2.5_ concentration in each city means for health, we calculated the exposure dose of PM_2.5_ (*Dose_PM_*_2.5_) amount, namely, how much PM_2.5_ penetrates our respiratory system.

First, we tried to calculate the reduced *Dose_PM_*_2.5_ (μg) for the 10-year-old children (*Dose_PM_*_2.5 10*-year-old children*_) per day and over two months after the city lockdown/self-reflection of each city. The calculation of the reduced *Dose_PM_*_2.5 10*-year-old children*_ was made by modifying the formula proposed by Löndahl et al. [25]
*Reduced Dose*_*PM*2.5 10-*year-old children*_ (μg) = *Reduced C*_*PM*2.5_ × *I/O ratio* × *F_dep._* × *T_exp._* × *R_bre._*(2)
where the *R**educed C_PM_*_2.5_ are 18.0, 5.3, and 0.36 µg/m^3^ in Wuhan, Daegu, and Tokyo, respectively, *I/O ratio* is the average indoor/outdoor (I/O) ratio of PM_2.5_ concentration at three different cities [26,27,28]. *F_dep_*_._ is the maximum deposition fraction at bronchiolar (Br.) Additionally, alveolar-interstitial (AI) regions suggested by Yamada et al. [29], *T_exp_*_._ is the exposure time (1-day or 60-day), and *R_bre._* is breathing rate (m^3^/h).

The *F_dep_*_._ and *R_bre._* are decided by the activity patterns of 10-year-old children. In this study, their daily activity patterns were classified into sleep, sitting or rest, light activities (exercise or movement), and heavy activities. The time allocated for each activity of the day is 9, 4, 10, and 1-h, respectively (see Table 1). Four kinds of their daily activity patterns were set up, assuming that they would have spent most of their time at home due to requests for self-reflection. 

The variables and the calculated reduced *Dose_PM_*_2.5_ (μg) for the 10-year-old children per day and over two months after the self-reflection of each city are summarized in Table 1.

In Table 1, the I/O ratios at Wuhan, Daegu, and Tokyo referred to Hua et al. (2017), Choi and Kang (2018), and Kagi (2014), respectively.

Figure 8 shows the reduced *Dose_PM_*_2.5 10*-year-old children*_ (μg) at Br. and AI regions over two months after the city lockdown/self-reflection of each city.

Wuhan, which had the largest decrease of PM_2.5_ concentration (18 μg/m^3^), also showed the largest reduced *Dose_PM_*_2.5 10*-year-old children*_ (3,660 μg at Br. and 6,222 μg at AI), followed by Daegu (445 μg at Br. and 1,287 μg at AI) and Tokyo (18 μg at Br. and 52 μg at AI) Additionally, the reduced *Dose_PM_*_2.5 10*-year-old children*_ (μg) varied greatly depending on the children’s behavior patterns. In all three cities, the reduced *Dose_PM_*_2.5 10*-year-old children*_ (μg) was high, in order of light activity > heavy activity > sleep > sitting/rest. The reason why the light activity was assessed to be a higher dose than heavy activity is because the activity time was set to be ten times higher. It is also clear that much more PM_2.5_ deposits at AI, a deeper part of the lungs, than in the bronchiolar.

### 3.3. Airway Inflammation Delay Effect

Through the thymic stromal lymphopoietin activation in mice, Liu et al. [30] suggested that the exposure to concentrations of PM_2.5_ equivalent to moderate pollution (31.6 µg of PM_2.5_) and severe pollution (100 µg of PM_2.5_) has been linked to the allergic airway inflammation (AAI) in mice.

It is very meaningful to calculate the amount of the exposure dose that can cause inflammation in humans (*Dose_Human_* (mg/kg)) indirectly through the experimental results with mice. Therefore, we tried to calculate the *Dose* for 10-year-old children (*Dose_PM_*_2.5 10*-year-old children*_ (mg/kg)) through the following equation, for the dose conversion from mouse to human introduced by Balakrishnan and Jacob [31].
*Dose*_PM2.5 10-*year-old children*_ (mg/kg) = *Dose_Mouse_*(mg/kg) × *K_m_* ratio(3)
where *K_m_* ratio = $\frac{K_{m\;Mouse}}{K_{m\;10-year-old\;children}}$.

Each *K_m_* i.e*., K_m Mouse_* and *K_m_*
_10*-year-old children*_ can be calculated with the following equation:(4)$$K_{m}=\frac{Weight\;\left(\mathrm{kg}\right)}{BSA\left(m^{2}\right)}$$

The *K_m_*
_10*-year-old children*_ for the Chinese, Korean, and Japanese were calculated by the average weight and body surface area (*BSA*) of boys and girls in each country. Their *BSA* were also calculated from following [32,33,34]: (5)$$\begin{array}{cc} BSA_{10-year-old\;children} & =0.008883\times {\mathrm{Weight}}^{0.444}\times {\mathrm{Height}}^{0.663}\\ & =0.007331\times {\mathrm{Weight}}^{0.425}\times {\mathrm{Height}}^{0.725}\\ & =0.00713989\times {\mathrm{Weight}}^{0.427}\times {\mathrm{Height}}^{0.516} \end{array}$$

Then, the *Dose_PM2.5_* for the AAI of 10-year-old children, i.e., the *AAI Dose_PM_*_2.5 10*-year-old children*_ (mg), can be calculated by following:
*AAI Dose*_*PM*2.5 10-*year-old children*_ (mg) = *Dose*_*PM*2.5 10-*year-old children*_ (mg/kg) × *Weight*_10-*year-old children*_ (kg)(6)

In this study, the *AAI Dose_PM_*_2.5 10*-year-old children*_ (mg) was calculated by the *Dose_Mouse_* (mg/kg), with 1.58 mg/kg (31.6 μg per mouse), on the assumption of medium air quality [30].

Additionally, the time (day) to reaching the *AAI Dose_PM_*_2.5 10*-year-old children*_ can be calculated by the following:(7)$$Day\;to\;reaching\;AAI\;Dose_{PM2.5\;10-year-old\;children}=\frac{AAI\;Dose{\;}_{PM2.5\;10-year-old\;children\;}}{Daily\;Dose_{\;PM2.5\;10-year-old\;children}}$$
where the *Daily Dose_PM_*_2.5 10*-year-old children*_ is the reduced *Dose_PM_*_2.5_ (μg) at the bronchiolar of the 10-year-old children per day during the city lockdown/self-reflection and is specified in Table 2.

Figure 9, the number of days it takes to cause allergic airway inflammation (AAI) by PM2.5 exposure, before and after the city lockdown/self-reflection at each city.

In the case of Wuhan, it took 25 days before the city lockdown, but 35 days after the city lockdown. Meanwhile, it was calculated that it took 130 to 164 days in Daegu and 570 to 587 days in Tokyo.

While it is easy to predict that inhaling clean air is good for children’s health, this study was able to quantitatively evaluate that the temporarily reduced PM_2.5_ concentration due to COVID-19 was effective for the delaying the AAI in three cities of Asia.

Meanwhile, there is still a possibility that much more harmful household air pollutants may have been exposed by staying indoors during the city lockdown/self-reflection periods.

## 4. Conclusions

In this study, the air quality variation was estimated in three cities in Asian countries experiencing the explosive outbreak of COVID-19, in a short period of time. The data assessment based on the actual measurements from the air pollution monitoring stations of each city clearly showed a quantitative reduction of PM_2.5_ and NO_2_. The health effect of the PM_2.5_ dose (reduced due to COVID-19) on 10-year-old children in each city was also quantitatively assessed. Especially, this study was able to quantitatively evaluate that the temporarily reduced PM_2.5_ concentration due to COVID-19 was effective for the delaying the AAI in three cities of Asia. There are many unmeasured factors other than PM_2.5_ level responsible for making the delay on AAI. Therefore, the delay effect of AAI estimated in the present study may be limited only by the effects of the reduced PM_2.5_. According to the study of Cohen et al. [10], 4.6 million people are dying annually because of the diseases and illnesses directly related to poor air quality. Therefore, the delay effect of AAI estimated in this study is limited only by the effects of the reduced PM_2.5_. According to the study of Cohen et al. [10], 4.6 million people are dying annually because of the diseases and illnesses directly related to poor air quality. Therefore, in terms of long-term human health hazards, the threat of air pollution can be much greater than that of COVID-19. Therefore, we should not only try to overcome the current situation of the COVID-19 pandemic, but at the same time seriously consider the new eco-lifestyle that we have to pursue after the end of COVID-19 pandemic.

## Figures and Tables

**Figure 1 ijerph-17-04119-f001:**
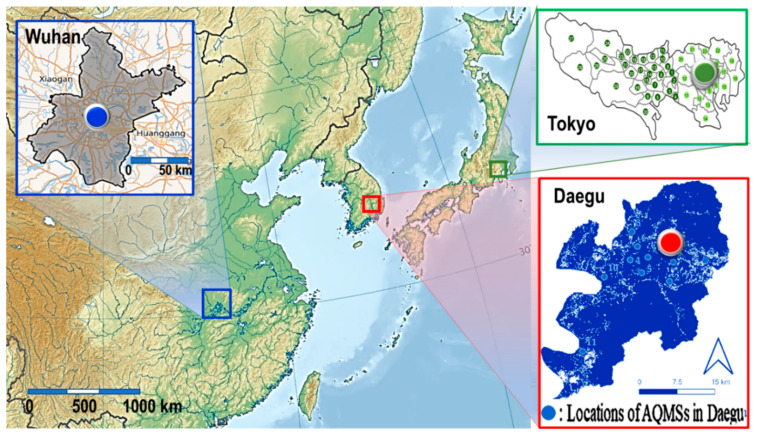
The maps of Wuhan, Daegu, and Tokyo and the locations of air pollution monitoring stations (AQMS) in each city.

**Figure 2 ijerph-17-04119-f002:**
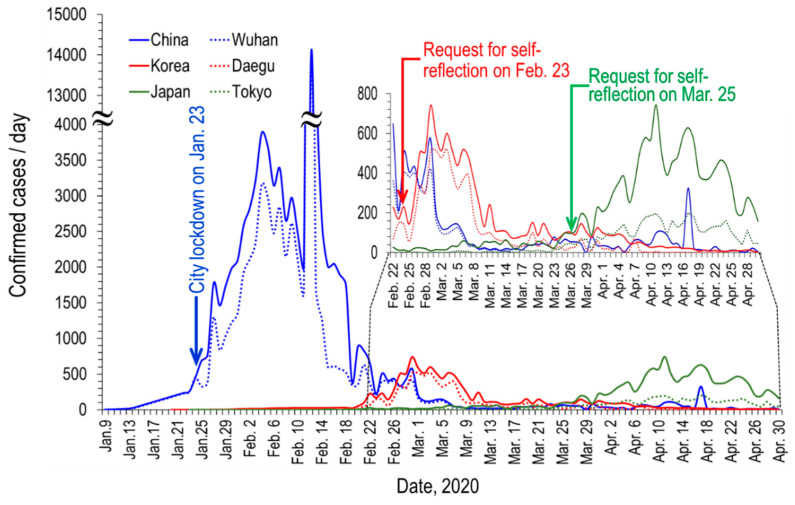
Timely variation of the confirmed cases of COVID-19 per day in Wuhan/China, Daegu/ Korea, and Tokyo/Japan.

**Figure 3 ijerph-17-04119-f003:**
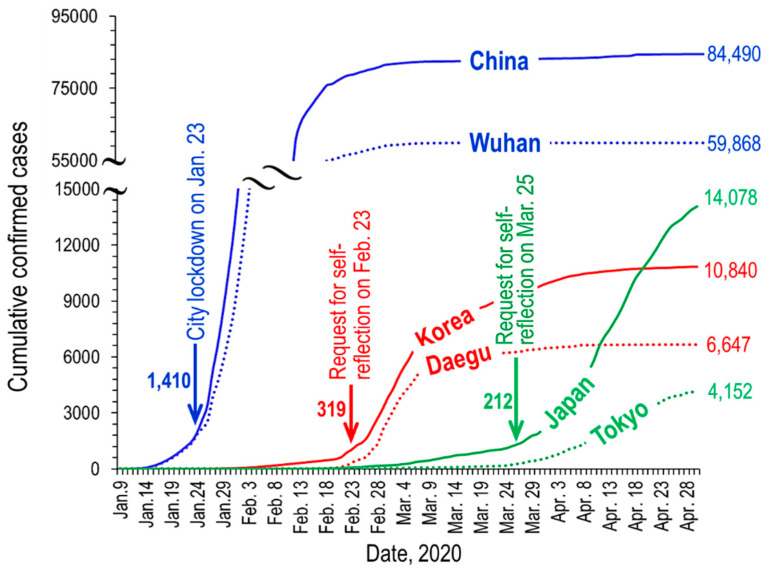
Timely variation of the cumulative status of COVID-19 in Wuhan/China, Daegu/Korea, and Tokyo/Japan.

**Figure 4 ijerph-17-04119-f004:**
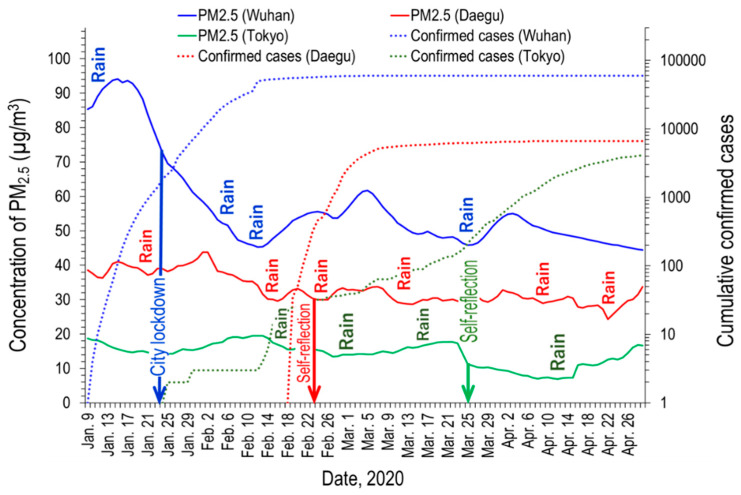
Daily variations of the concentration of PM_2.5_ in Wuhan, Daegu, and Tokyo, with the cumulative confirmed cases of COVID-19.

**Figure 5 ijerph-17-04119-f005:**
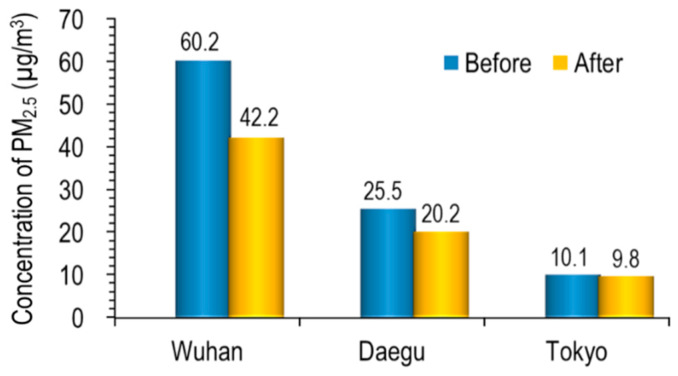
Comparison of PM_2.5_ concentration before and after each city’s self-reflection.

**Figure 6 ijerph-17-04119-f006:**
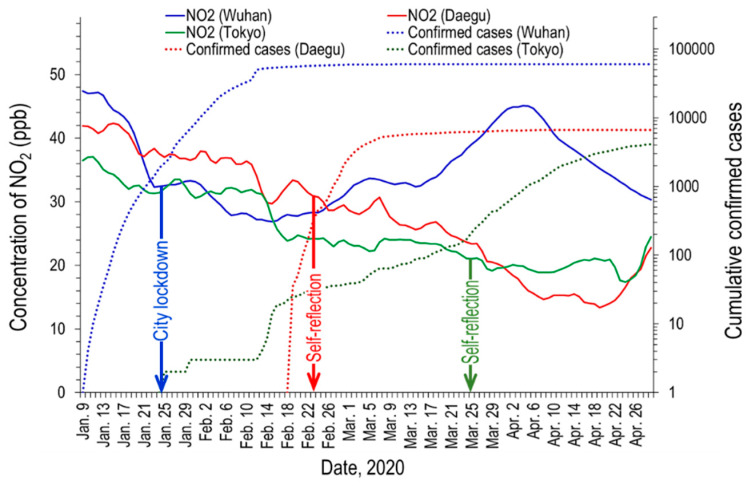
Daily variations of the concentration of NO_2_ in Wuhan, Daegu, and Tokyo, with cumulative confirmed cases of COVID-19.

**Figure 7 ijerph-17-04119-f007:**
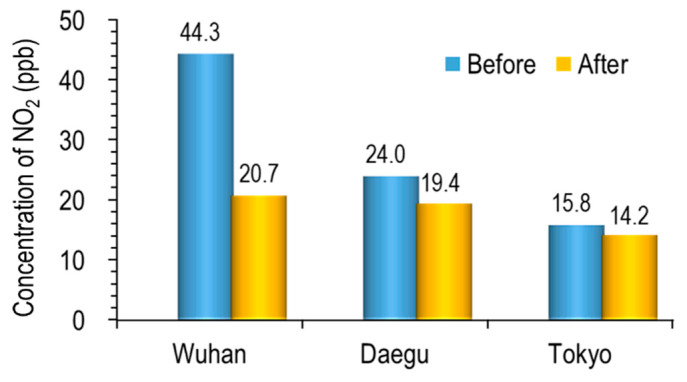
Comparison of NO_2_ concentration before and after each city’s self-reflection.

**Figure 8 ijerph-17-04119-f008:**
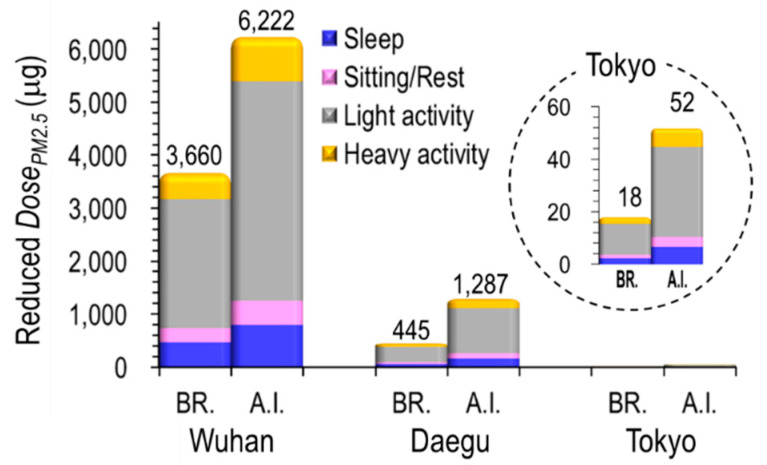
The reduced *Dose_PM_*_2.5 10*-year-old children*_ (μg) at bronchiolar and alveolar-interstitial (AI) region over two months after the self-reflection of each city.

**Figure 9 ijerph-17-04119-f009:**
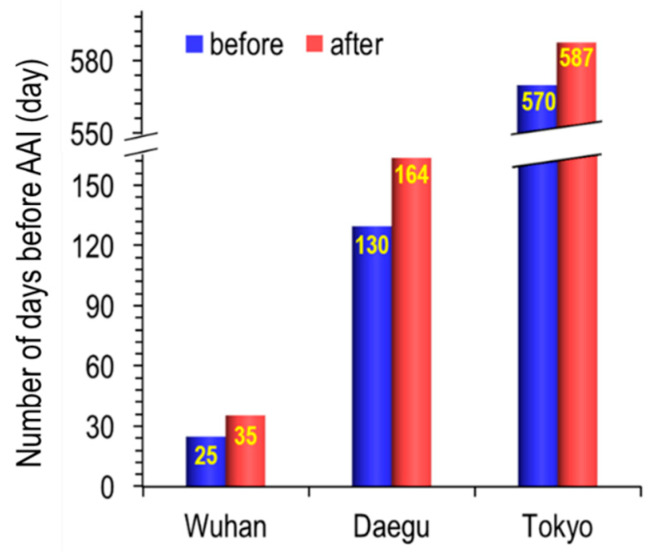
The number of days it takes to cause AAI by PM_2.5_ exposure, before and after the city lockdown/self-reflection at each city.

**Table 1 ijerph-17-04119-t001:** The variables for calculation of the reduced *Dose_PM_*_2.5_ (μg) for the 10-year-old children per day and over two months after the self-reflection of each city.

	Behavioral Patterns of 10- Year-Old Children in the Day	Activity Time (h)	Total Exposure Period (*T_exp_.*, Day)	*C_PM2.5_* (μg/m^3^) Reduced in 2020	I/O Ratio	*F_dep._*		*R_bre_*. (m^3^/h)
						Br.	A.I.	
Wuhan	Sleep	9	60	18	0.94	0.209	0.355	0.246
	Sitting/Rest	4	60	18	0.94	0.218	0.370	0.301
	Light activity	10	60	18	0.94	0.270	0.459	0.888
	Heavy activity	1	60	18	0.94	0.300	0.510	1.610
Daegu	Sleep	9	60	5.3	0.66	0.123	0.355	0.246
	Sitting/Rest	4	60	5.3	0.66	0.128	0.370	0.301
	Light activity	10	60	5.3	0.66	0.159	0.459	0.888
	Heavy activity	1	60	5.3	0.66	0.176	0.510	1.610
Tokyo	Sleep	9	60	0.36	0.39	0.123	0.355	0.246
	Sitting/Rest	4	60	0.36	0.39	0.128	0.370	0.301
	Light activity	10	60	0.36	0.39	0.159	0.459	0.888
	Heavy activity	1	60	0.36	0.39	0.176	0.510	1.610

**Table 2 ijerph-17-04119-t002:** Reduced *Dose_PM_*_2.5_ (μg) at the bronchiolar and AI regions of the 10-year-old children per day and over two months after the city lockdown/self-reflection of each city.

	Behavioral Patterns of 10- Year-Old Children in the Day	Activity Time (h)	*Reduced Dose_PM2.5_* (μg) at Br.		*Reduced Dose_PM2.5_* (μg) at A.I.	
			1 Day	2-Month	1 Day	2-Month
Wuhan	Sleep	9	7.82	469	13	798
	Sitting/Rest	4	4.43	266	8	452
	Light activity	10	40.57	2434	69	4138
	Heavy activity	1	8.17	490	14	834
	Total reduced *Dose_PM2.5_* (μg)		61	3660	104	6222
Daegu	Sleep	9	0.95	57	3	165
	Sitting/Rest	4	0.54	32	2	93
	Light activity	10	4.93	296	14	855
	Heavy activity	1	0.99	60	3	172
	Total reduced *Dose_PM2.5_* (μg)		7	445	21	1286
Tokyo	Sleep	9	0.04	2.3	0.1	7
	Sitting/Rest	4	0.02	1.3	0.1	4
	Light activity	10	0.20	11.9	0.6	34
	Heavy activity	1	0.04	2.4	0.1	7
	Total reduced *Dose_PM2.5_* (μg)		0.30	17.9	0.9	52

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
