# Peer review of "Air Quality Variation in Wuhan, Daegu, and Tokyo during the Explosive Outbreak of COVID-19 and Its Health Effects"

_ijerph, 2020, doi:10.3390/ijerph17114119_

Round 1

Reviewer 1 Report

This study was designed to evaluate and compare the variations of air quality during the outbreak of Covid-19 and its health effects, in Wuhan, Daegu and Tokyo, with a focus on the average PM 2.5 of one month before and after the lockdown (Wuhan) and self -reflexion (Daegu and Tokyo), in the bronchiolar and alveolar areas of the 10-year-old children.

This study is very important because it shows several qualities and the results are clear and useful for physiscians and public helath deciders:

-he study is performed in 3 big cities in 3 different countries providing results that can be generalized on several continents

-the methodology to count the Covid-19 cases, the measure of PM2.5 and NO2 , were standardized

-the results are clear:PM2.5 levels was decreased by 29.9,20.9 and 3.6 in Wuhan, Daegu and Tokyo respectively,when the 3 cities in this order also recorded 53.2,19.0 AND 10.4 % fall of NO2 levels.Wuhan that developed an official lockdown had the largest decrease of PM2.5, also had the largest reduced DosePM2.5 10-year-old children at the bronchiolar levels , followed by the 2 cities in self -reflexion (Daegu and Tokyo ) over 2 months after the city lockdown/self -reflection.

These results validate the positive impact of lockdown/self-reflection on air pollution and airway inflammation.

-the Figures and Tables are very useful and the references are complete

But one major problem is observed:

-the plan is not usual and confuse,so the analyse of the article is difficult.Example:thechapter Discussion is associated tothe chapter Results, the descirption of the methods of Airway Inflammation Delay Effect is reported in the chapter Results and Discussion

-English text must be reviewed

Author Response

The plan is not usual and confuse, so the analysis of the article is difficult.

Example:

the chapter Discussion is associated to the chapter Results, the description of the methods of Airway Inflammation Delay Effect is reported in the chapter Results and Discussion

The authors think it would be more reasonable not to move the estimation method of “Airway Inflammation Delay Effect” to the chapter of Method because it describes not the simple calculation method, but also the newly proposed method in this study.

Meanwhile, since the “simple moving average (SMA)” is the simple data handling, it moved to 2.3. Data handling.

-English text must be reviewed

The correction of English has been made.

Reviewer 2 Report

The topics covered in this draft are considered timely. By comparing and evaluating the situation of air quality reduction due to this COVID-19 pandemic by each city in three countries, it will be a useful resource for devising policy measures for future air quality reduction. However, regarding the health effects measured by the delayed days to get AAI, authors need to reconstruct the conclusion mentioned. Their estimates on the delayed period on AAI seem to be simply a function of the ambient level of PM2.5. Other factors influencing possibly to those estimates seem to be considered as a constant, which is very unrealistic and impractical. There are many unmeasured factors other than PM2.5 level responsible for making this delay on AAI, although we will really observe the delayed days after the self-reflection or whatever. Therefore, it can be suggested that the draft should be limited to the issue of air quality improvement. 

Author Response

  1. However, regarding the health effects measured by the delayed days to get AAI, authors need to reconstruct the conclusion mentioned.

As commented, below was newly added into the conclusion of the revised manuscript.

Especially, this study was able to quantitatively evaluate that the temporarily reduced PM2.5 concentration due to COVID-19 was effective for the delaying the AAI in three cities of Asia.

  1. Their estimates on the delayed period on AAI seem to be simply a function of the ambient level of PM2.5. Other factors influencing possibly to those estimates seem to be considered as a constant, which is very unrealistic and impractical. There are many unmeasured factors other than PM2.5 level responsible for making this delay on AAI, although we will really observe the delayed days after the self-reflection or whatever. Therefore, it can be suggested that the draft should be limited to the issue of air quality improvement.

As many published papers below mentioned, it is well-known fact that the inflammation of various respiratory diseases are associated with PM2.5 inhalation. Children are particularly vulnerable to air pollutants because their immune and antioxidant defence mechanisms are still developing and they have a faster breathing rate, taking in more air per unit body weight than adults, resulting in inhalation of higher doses of air pollutants compared with adults.

According to Stanley (2015), inflammation may occur as a result of either immunologic or nonimmunologic airway insults, which produce airway edema and cause the immigration of inflammatory cells into the lumen through the epithelium.

As pointed out by a reviewer, many factors may be involved in inflammation, including one that has not yet been clarified.

Therefore, below was newly added into Conclusion of the revised manuscript.

There are many unmeasured factors other than PM2.5 level responsible for making this delay on AAI. Therefore, the delay effect of AAI estimated in the present study may be limited only by the effects of the reduced PM2.5.

Kampa M, Castanas E. Human health effects of air pollution. Environ Poll 2008;151:362–7.

Mannucci PM, et al. Effects on health of air pollution: a narrative review. Intern Emerg Med 2015;10:657–62.

Huang SK, et al. Mechanistic impact of outdoor air pollution on asthma and allergic diseases. J Thorac Dis 2015;7:23–33.

Araujo JA, Nel AE. Particulate matter and atherosclerosis: role of particle size, composition and oxidative stress. Part Fibre Toxicol 2009;6:24.

Environmental Protection Agency. Particle pollution and your health. Available at: https://nepis.epa.gov/Exe/ZyPDF.cgi?Dockey=P1001EX6.txt. Last accessed: 22 July 2018.

Achilleos S, et al. Acute effects of fine particulate matter constituents on mortality: a systematic review and meta-regression analysis. Environ Int 2017;109:89–100.

Stanley F. Malamed DDS, ... Daniel L. OrrII DDS, MS (ANES), PHD, JD, MD, in Medical Emergencies in the Dental Office (Seventh Edition), 2015

Reviewer 3 Report

My opinion is major revesion

Comments:

The research on air quality improvement and health benefits during the COVID-19 is a meaningful work. In this paper, the air quality and health effect in three cities (Wuhan, Daegu, and Tokyo) in Asian countries have been discussed. The reduction of air pollutants and DosePM2.5 at the bronchiolar and 12 alveolar-interstitial regions have been reported. Some suggestions have been given as follow:

In this article, the study periods have been divided into different stages, like lockdown stage, self-reflection stage, a table is needed to show the division of time periods.

In line 121-126, the 5-day simple moving average has been introduced, this part should be put in the Methods.

In line 132-139, the reduction caused by city lockdown or seasonal reductions have been discussed, whether the relative contribution can be further discussed?

In line 174-175: from the reference 19, we do not find the reduction of NO2 during COVID-19 epidemic saved 100,000 lives in China.

In line 219, how to calculate the Airway Inflammation Delay Effect should be given. And the delay effect for Airway Inflammation caused by lockdown /self-reflection should be discussed.

Author Response

  1. In this article, the study periods have been divided into different stages, like lockdown stage, self-reflection stage, a table is needed to show the division of time periods.

In this study, the period of study was not divided into three periods, but the difference in air quality before and after the time each city implemented the Corona measures (i.e., the city lockdown in Wuhan and self-reflection in Daegu and Tokyo) was discussed.

Data estimation for the three cities was also conducted for the entire period of the study, namely, from January 9 to the end of April.

Instead, the measures for the COVID-19 at each city (i.e., the city lockdown in Wuhan and self-reflection in Daegu and Tokyo) were marked in Figures 2, 3, and 4, which show time series data.

  1. In line 121-126, the 5-day simple moving average has been introduced, this part should be put in the Methods.

Yes, as commented, we moved its position to Methods.

  1. In line 132-139, the reduction caused by city lockdown or seasonal reductions have been discussed, whether the relative contribution can be further discussed?

Further comparison with the results of 2017 (Wu et al.) was made to clarify the reduction of PM2.5 due to the city lockdown of Wuhan.

In February, when the number of the confirmed COVID-19 cases was at its peak, PM2.5 concentration was 57.3% lower than that in February, 2017 (121.2 μg/m3) [18]. Therefore, it can be said clearly that the city lockdown has resulted in a great reduction in PM2.5.

[18] Wu, X.; Wang, Y.; He, S.; Wu, Z. PM2.5 /PM10 ratio prediction based on a long short-term memory neural network in Wuhan, China. Geosci. Model Dev., 2020, 13, 1499–1511.

  1. In line 174-175: from the reference 19, we do not find the reduction of NO2 during COVID-19 epidemic saved 100,000 lives in China.

It has revised as below with a new reference;

The model calculation by Dutheil et al. [19] suggested that the reduction of NO2 in China due to COVID-19 epidemic during a time period of two months saved around 100,000 lives in China.

  1. Dutheil, F.; Baker, J.S.; Navel, V. COVID-19 as a factor influencing air pollution? Environ Pollut. 2020, 263(Pt A), 114466.

In line 219, how to calculate the Airway Inflammation Delay Effect should be given. And the delay effect for Airway Inflammation caused by lockdown /self-reflection should be discussed.

Additionally, the time (day) to reaching the AAI Dose PM2.5 10-year-old children can be calculated by following:

eq. cannot appeared here.

where the Daily DosePM2.5 10-year-old children is the reduced DosePM2.5 (mg) at the bronchiolar of the 10-year-old children per a day during the city lockdown/self-reflection and it specified in Table 2.

While it is easy to predict that inhaling clean air is good for children's health, this study was able to quantitatively evaluate that the temporarily reduced PM2.5 concentration due to COVID-19 was effective for the delaying the AAI in three cities of Asia.

Meanwhile, there is still a possibility that much more harmful household air pollutants may have been exposed by staying indoors during the city lockdown/self-reflection periods.

Round 2

Reviewer 1 Report

The article is improved but the Results and Discussion remain mixed and are not separated.

The article remains very interesting

ok to publish it

Author Response

Thank you for your sincere review of our paper, and we will consider again in the process of publishing whether we will separate the discussion from the results.

Reviewer 2 Report

Authors argued that they evaluated the effect of PM reduction on the delay of AAI in three different cities. This must not be an observed finding, but a further research hypothesis to be tested. There are many epidemiological findings that provide a positive relationships between PM level and adverse health outcomes, including mortality, hospital admissions due to cardio-pulmonary diseases, and some of selected inflammatory markers, etc. With all these findings, we may propose the possible health benefits including the delayed AAI due to PM reduction, although we have not confirmed those benefits yet. The estimated result on the AAI delay should be mentioned only at the Discussion section and can be treated as a main finding of this manuscript. 

We all expect to see the health beneficial effects including the AAI delay from this unusual situation generated by the pandemic event. These hypotheses would be confirmed and validated sooner or later with the real outcome dataset. The one of the challenging parts is to make sure how much the health beneficiary effect (if we observe it with the real dataset) is due to PM reduction. Simply, it is quite true for this pandemic situation to reduce the ambient PM level and change our daily lives to stay longer time at indoor environment by social distancing so that PM exposure would be decreased as well. There are many other factors we should consider for measuring the associations. 

In conclusion, I suggest that authors should present one main finding on PM reduction regarding this pandemic situation and describe the estimation on AAI delay at the Discussion section. I also recommend, therefore, the title of the manuscript would be changed.

Author Response

Thank you for your sincere review of the dissertation, and I will refer to the opinions of other reviewers about changing the title of the paper.

Reviewer 3 Report

no

Author Response

Thank you for your sincere review of our manuscript.